# An Analytical Survey of WSNs Integration with Cloud and Fog Computing

Qaisar Shaheen [1,*], Muhammad Shiraz [2,3], Shariq Aziz Butt [4], Abdullah Gani [5,*] and Muazzam A. Khan [6]

1 Department of Computer Science & Information Technology, Superior University, Lahore 54000, Pakistan
2 Department of Computer Science, Federal Urdu University of Arts, Science & Technology, Islamabad 44000, Pakistan; drmuhammadshiraz@fuuastisb.edu.pk
3 Department of Computer Science, Allama Iqbal Open University, Islamabad 44000, Pakistan
4 Department of IT and CS, The University of Lahore, Lahore 54000, Pakistan; Shariq2315@gmail.com
5 Faculty of Computing and Informatics, University Malaysia Sabah, Jalan UMS, Kota Kinabalu 88400, Malaysia
6 Department of Computer Sciences, Quaid-i-Azam University, Islamabad 44000, Pakistan; khattakmuazzam@gmail.com
* Correspondence: qaisar.shaheen2002@gmail.com (Q.S.); abdullahgani@ums.edu.my (A.G.)

**Abstract:** Wireless sensor networks (WSNs) are spatially scattered networks equipped with an extensive number of nodes to check and record different ecological states such as humidity, temperature, pressure, and lightning states. WSN network provides different services to a client such as monitoring, detection, and runtime decision-making against events occurrence. However, the WSN network still has some limitations in computing power, storage resources, and battery life, which make the network is restricted for data transformation. It is due to less supportive battery power, and limited memory of nodes. The integration of WSN and cloud offers an open, adaptable, and more reconfigurable stage for different security checks and regulating requirements. In this paper, we discovered how WSN and cloud computing (CC) are integrated and help to accomplish different goals. Additionally, a comprehensive study about procedures and issues for an effective combination of WSN-CC is presented. This work also presents the work proposed by the research community for WSN-CC. Besides, we explored the integration of WSN/IoT with Fog computing (FC). Based on investigations, WSN integration with Fog computing (FC) has many benefits with respect to latency, energy consumption, data processing, and real-time data streaming. FC is not a substitute for distributed computing, so far it is utilized to improve the productivity of the sensor.

**Keywords:** fog computing; cloud computing; WSN to fog; WSN to cloud

## 1. Introduction

Mostly there are two kinds of networks, wired and wireless. The wireless sensor network (WSN) is the most used network for the connectivity of devices and communication. The WSN network is applied in many daily-based applications such as environment monitoring where humans cannot reach, health monitoring of patients using the WSN, industrial monitoring, and air pollution monitoring. WSN spatially scattered sensor networks interconnect with sensed data in these situations. Regardless of the various uses of WSN network and privilege of easy connectivity of devices, the network has some limitations such as data processing sensed by the sensors deployed in the environment, temporary storage of data when a large number of sensors are deployed in the environment, tools and software use, low battery power of sensors, and sensor's integration in a single platform. Additionally, the WSN middleware applications are to address the gap around high-level specifications. Several other problems need to be addressed for applications and the difficulty of the operations in the underlying network. Due to these limitations with the WSN network the cloud computing (CC) plays a significant role in the network. The

integration of WSNs and clouds can also be used in a large number of applications such as transportation, war zones, health, and agriculture [1,2]. Disaster surveillance is another region, in which sensor nodes can be used to recognize the tragedy by exact investigated points, to decrease the causality and damage of property. Cloud computing has a slightly positive impact on WSN in the following ways: integration of sensors, ease of storage of data on the cloud, ease of data processing, easy accessibility of tools in different WSN environments, load balancing of a network by the CC, and CC need for WSN to develop other similar computing models due to its low cost. All of the WSN network limitations can be overcome by the CC placement. The WSN with cloud integration is shown in Figure 1. The emergence of WSN and cloud computing services has introduced significant sensor-cloud integration opportunities that will make it easier for users not only to track their objects of concern via sensors but also to employ cloud services to evaluate future directions [3].

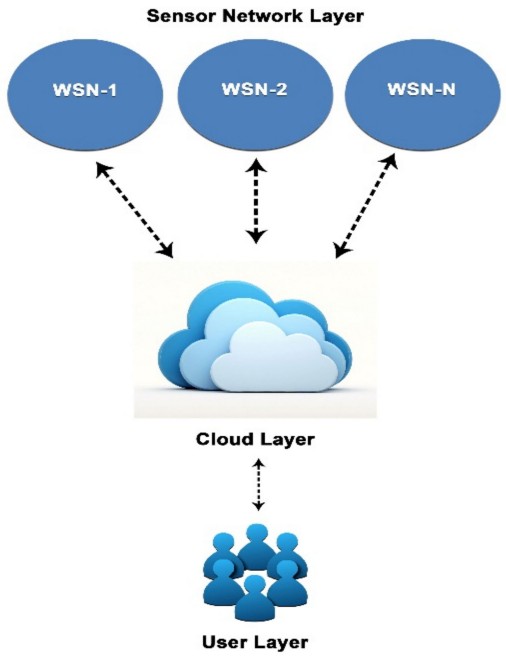

**Figure 1.** WSN to cloud integration.

Integration of WSN with the cloud can also be achieved through FC. WSN to fog integration is shown in Figure 2. The figure explains the concept of fog working in the WSN [4]. Different clients can use the Fog as different services for servers to perform their activities. Fog computing provides the smart data processing of WSN network sensors. For example, the goal is to reduce sending direct sensor information toward the cloud thus improving the ratio for both user data and noise [5]. Some basic information processing algorithms are introduced at the sensor stage [6,7]. Fog is a three-tier structure which is shown in Figure 3. Fog maximizes throughput and minimizes the latency for energy saving [8–10]. Fog is a very flexible structure for providing services to cloud and sensing nodes [11–13].

This integration is based on a two-tier structure, and Figure 3 elaborates this clearly. In this paper, we clearly define the WSNs integration with cloud computing and with Fog computing. The significance of the study is to illustrate the key benefits, issues, and introduced frameworks, techniques in this combination. This study discusses the Fog and cloud requirements as well as the integration of WSN with the Fog and cloud. With, We discuss this in detail next in Section 2.

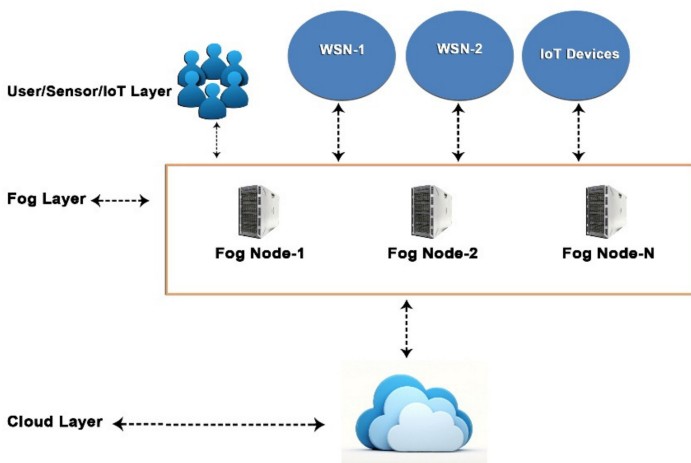

**Figure 2.** WSN to Fog integration.

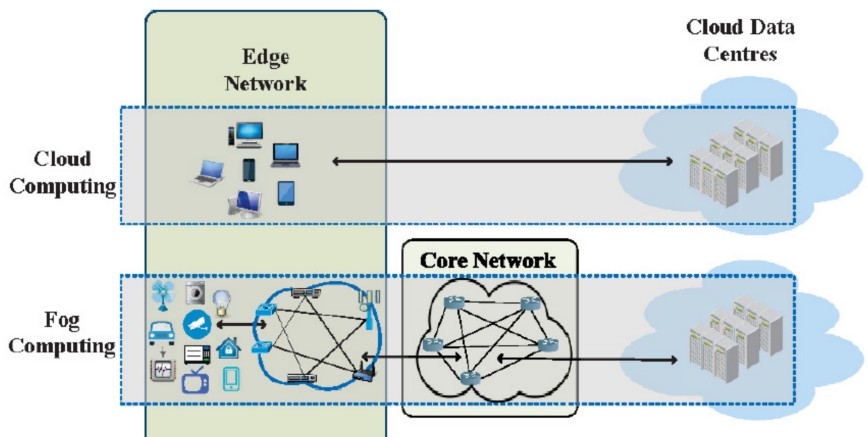

**Figure 3.** Fog computing vs cloud computing architecture.

## 2. Background of Study

For this great combination, there is a need to describe the requirements for a system. From this line of research, we explain these requirements in Table 1 for cloud and FC. CC is a rising technology for the modern era that provides services to users through the Internet. CC data and applications are placed at some shared locations on the Internet and servers are placed at some remote locations which are accessed by users through the Internet. It also permits resource sharing by reducing space and cost. Considering the popularity of the cloud and its advantages, people are shifting to cloud services progressively. Many cloud service providers provide services to users. CC services give more benefits as compared to a conventional computing paradigm. These benefits include reliability, strategic edge, and manageability, and most importantly low cost. It is easier for users to access their data from anywhere by using CC irrespective of the place and machine, as the data are located in the central location [14,15]. The main purpose that all cloud service providers seek to offer the finest cloud services is that they are competing with time to make it better with every passing day. A hybrid cloud may be an association for both people in general and private. A community cloud setup is deployed by a community to achieve its objectives. Cloud provides services in three fundamental categories: Infrastructure as a Service (IaaS), Platform as a Service (PaaS), and Software as a Service (SaaS) [16,17]. IaaS is a model that combines two parties' customers and cloud providers. It plans for the virtual delivery of cloud resources at the client's doorstep. Customers can control the resources as per their needs. The cloud provider extends benefits for customers by paying for storage space, processing, and network. PaaS enables clients to use the operating system as a service.

The same required operating system can be rented to numerous clients. Through SaaS, clients can use the software to rent/service [18,19]. The rest of the paper is divided into six segments. In segment 2 of the paper, CC foundation and evolution are introduced. In segment 3, we discuss related work. In segment 4, a comparison of different proposed integration frameworks of WSN with cloud and Fog is presented [20,21]. In segment 5, the open issues are identified with the understanding of the WSN and cloud gaps, and future work of this domain is highlighted. In segment 6, the present review is concluded briefly [22–24].

**Table 1.** Requirements for cloud and FC.

| Requirement | Cloud Computing | Fog Computing |
| --- | --- | --- |
| Latency | High | Low |
| Connectivity | Dedicated line | Wireless |
| Service location | In the Internet | Edge of the network |
| Hops between server and client | Multiple | One |
| attacks on en-route data | Higher probability | Very low probability |
| Location awareness | No | Yes |
| Mobility | Limited | Fully supported |
| Interaction for real-time | Supported | Supported |

*Cloud Computing Evolution*

Internet usage is increasing rapidly day by day, which plays a vital role in the evolution of CC. There is a substantial shift of people to the Internet because a lot of devices are available. People can access web services from anywhere through mobile phones, laptops, and desktops. It has become an essential part of their lives [25,26]. Nowadays, the internet has become one of the biggest sources of information about education, health services, entertainment, and many other daily life issues. This is the reason that the web has modernized or started using new technology for information sharing. Over time, the vast usage of the Internet leads to the invention of new things in Internet technology. Due to innovation on the internet, CC is emerging rapidly. CC is a prominent emerging technology of the present day that has its roots back in the 1950s when mainframe computers came into the information technology (IT) industry. Mainframe computers caused the birth of the cloud by going through enterprise transformation. The cost of a mainframe computer was so high that companies were not financially strong enough to buy the standalone device. Multiple users and companies used to share mainframe devices. In this way, the concept of shared resources took place in the information technology industry, in which multiple companies were using the same mainframe device through terminals to save cost. Through the concept of shared resources, cost-saving was the biggest advantage of that time and motivated the researchers and IT people to start thinking about it [27,28]. In the 1970s, a virtual machine (VM) having an operating system was launched by International Business machines (IBM) that presented the concept of visualization in computing. More than one operating system could be run simultaneously on one machine. In this concept, more than one operating system that can be named as the guest operating system runs on the same machine for sharing resources. At this point, resource sharing was one more feature that motivated the researcher to do work and introduced the new things in this field for better utilization of resources by saving cost and time [29,30].

**3. Related Work**

In this section, we extensively review the related works on Fog computing and cloud interaction. All the related studies are about Fog and cloud data accessibility, data storage, safety, connectivity, and location awareness and all these features are stated in Table 1. All

these are the Fog and cloud computing requirements. By considering all these requirements we enlisted only those frameworks that suggested according to such requirements.

Fog structure is a three-layer system in which a different portable sink is proposed. The system comprises a sink layer, a fog layer, and a directing layer. In the sink layer, every sink position is utilized as a generator to get the Voronoi graph. At that point, in the fog layer, the sinks go as fog nodes, and each sink coordinates [31,32].

The execution of Fog-to-Cloud structures demonstrates the advantages of organizing the diverse limits brought by the two devices at the cloud and the edge. With that in mind, the paper centers on the DSE issue. Two techniques; First-Fit and Random-Fit are also proposed for resource allocation [33,34]. A service delay minimizes the use of IoT fog-cloud applications. To overcome this issue a study proposed a framework [35,36]. A fog IFCIoT framework that guarantees expanded execution, vitality, productivity, and versatility, for IoT and CPS applications has been proposed in [37,38]. This work discussed the objectives and difficulties of the fog platform. They have presented the strategies and execution of a prototyping stage for fog processing. Finally, they have assessed their prototyping phase in Smart Home applications [39,40].

In a cloud-based framework, face identification and resolution to access personal information or data are proposed. At that point, a parallel coordinating system and distributed computing-based determination structure are proposed to effectively resolute confronted persons, control individual information, and secure a person's data [41,42].

A FOGG framework is developed for sensor network integration with the internet in [43,44]. In this framework, the authors used a dedicated device to work as an IoT gateway with additional services, security controller functionality, and protocol translation. ICN security feature is used in FOGG to provide data-centric instead of communication. In [45], the author proposed a new Stable Election Protocol (N-SEP) that elects the best cluster head based on distance from the base station, heterogeneity ratio, and energy consumption. This framework provides 50% and 25% stability to LEACH and SEP, respectively, in terms of network lifetime and energy efficiency [46].

WSN framework empowered for localization and activity monitoring in the context of Ambient Assisted Living AAL for real-time activity through the edge mining approach. For cloud base analysis, the genetic algorithm is used. The proposed system achieved high positional accuracy and low frequency of data computation on sensor devices [47,48].

In Fog-based IoT applications, a Distributed Dataflow programming model is introduced in this paper [49]. They examine the center prerequisites, that is, Fog-based IoT applications distinguish various problems with existing ways to deal with Fog-based application improvement. A fog framework-based information obtaining an instrument, which highlights lightweight, high productivity, and constant strangely information separating calculation is proposed in [50]. Through community-oriented figuring of all WSN nodes, the approach empowers solid and proficient information securing even if there should arise an occurrence of information variation from the norm [51].

The FC layer in WSN is introduced as a response to the issues exhibited in the investigation. The temperature measurement database managed through Link Quality Routing Protocol (LQRP) is used which is based on Ad-hoc On-Demand Distance Vector (AODV) [52]. The work centers on demonstrating and investigating the association of the Fog paradigm with the plan of a CoT (Cloud of Things) foundation. The objective of the paper is to build a scientific model of three-level CoT to evaluate the relevance of the fog level with regards to the framework and to show that it is a key factor to meet the requests of time-limitation applications [53].

An adaptable processing framework that empowers more brilliant urban communities with mobile data through enormous information innovation is proposed in [54,55]. An IoT framework and a door that enable applications to be sent nearer to the system edge and relocated to the cloud in light of the client's necessity are proposed in [55].

They focused on the identification of unanticipated sensor data collected by the various sensors or the surroundings during investigation in this paper. We offer a new method for

automatically detecting anomalies in heterogeneous sensor networks that combine edge and cloud analysis of the data [56].

This topic is investigated in this work. We begin by defining the concept of scope in a multi-IoT scenario. Then, we present two formalizations of this concept that allow its values to be computed. After that, they go through two different ways and that scope can be used. Furthermore, they present a set of tests that are conducted to assess it; the final one compares range to dispersion degrees and impact degree, two characteristics that have previously been suggested in the research [57,58].

The idea recommended in [59,60] included the use of wireless sensors to detect the nearness of traffic close to any circle or common area and then made the traffic path easy and accessible. This development does not require any framework in vehicles so can be executed in a traffic system affordably and effectively in less time. A comparison of fog and CC is given in Table 2.

**Table 2.** Rule-based recommendations for logs.

| Domain | Cloud Computing | Fog Computing |
|---|---|---|
| Network requirement | High-speed bandwidth, High-speed servers. | Do not need high speed, any the device can act as a Fog |
| Applications | A required special type of application and these applications mainly suffer high latency. | Can be used in critical applications because of very low latency. |
| Operations/management | Required special team to manage and operate in a fully controlled environment. | Operate as per user need in their environment and not operated by a person. Can manage any type of company. |
| Deployment | Required special planning for deployment. | Do not need special planning mostly required intense planning. |
| Size | High cloud needs a large size of network with at least thousands of servers. | Size of a network as per user need and every fog node can be a single server. |
| Network model | Centralized | Distributed and scattered in geo-graphical area. |
| Location/space | Small | Scattered in a geographical area. |
| Scalability | Scalable at center | Scalable for both center and fog |

## 4. Research Methodology

The objective of the research is to integrate WSN with the Fog and cloud computing. We find the existing works related to our scope of study from the reputed search engines like IEEE, Springer, Taylor and Francis, Wiley & Sons, ELSEVIER, and MDPI. We searched papers with the following keywords:

- WSN combinations with areas.
- WSN combination with Fog computing.
- WSN combination with cloud computing.
- WSN survey study with cloud and fog computing.
- Fog computing application with WSN.
- Fog computing with WSN future directions.
- WSN architectures for Fog and cloud computing.
- Survey of Fog and cloud computing.

Our adopted research methodology is presented in Figure 4 with the sections that we used in our study. These sections are mostly used for the literature work.

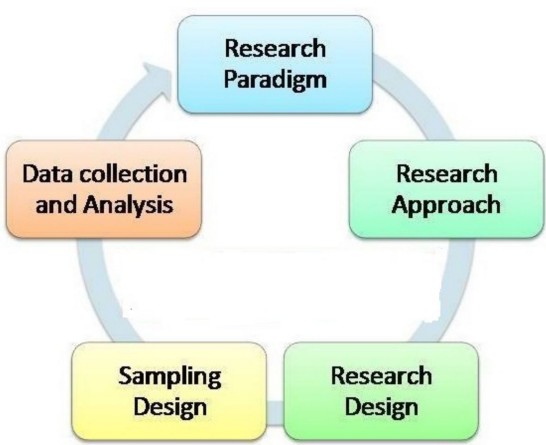

**Figure 4.** Adopted research methodology.

We conducted this survey study, by adopting a research methodology with the different components. These research components that we adopted are listed as follows:

### 4.1. Research Paradigm

The research approach is the descriptive research followed for a research study, suggested by the research group.

### 4.2. Research Approach

The research methodology specifies the requirements for collecting knowledge, interpretation, and explanations. In research, we implemented survey data gathering.

### 4.3. Research Design

It is the structure of approaches and techniques used in collecting and analyzing the proportions of the variables found in the difficult study.

### 4.4. Sampling Design

The sample contains the individuals in your survey as the outcome of the interview or questionnaire, but we have no individuals in our research.

### 4.5. Data Collection and Analysis

It involves the processing and analysis of data by using the methodology of the predefined theory. To identify the research papers applicable to our research, we followed these approaches. We perform research on systematic research on the WSN combination with the Fog and cloud computing. In this study, we reveal that how this combination has a great influence on technology. How this combination impacts different domains of research. How this combination provides different facilities to users. This area of research already has the attention of the research community, i.e., introducing new studies for the development of WSN with Fog and cloud computing. From this line of research, we draw some research questions that are adopted during the complete study, as follows:

**RQ1:** What is WSN with Fog and cloud computing's impact in IoT?
**RQ2:** What are technologies done by the research community?
**RQ3:** What are the benefits of WSN with Fog and cloud computing for users?

Our entire strategy for study methodology relies on search engines such as Google Scholar, Scopus, and Google. A systematic literature review provided numerous elements that are significant in defining the scope of the study. The rest of the elements are related to entities, methods, and processes. The functional dimensions have a significant influence on the technological context plan scope. The method is designed for the division of elements

into parts and sections by CII with categories. For the research review, we used various factors as stated in Table 3. Table 4 presents the searched key strings.

**Table 3.** Occurrence of factors in the literature.

| Elements | Occurrence in Literature |
|---|---|
| WSN combinations with areas. | 1, 2, 4, 6, 7, 22, 23, 25, 26, 28 |
| WSN combination with Fog computing. | 3, 5, 8, 9, 10, 35, 38, 39, 40, 42, 45, 46 |
| WSN combination with cloud computing. | 1, 2, 11, 13, 15, 27, 25, 22, 29, 30, 35, 48, 50, 51 |
| WSN survey study with cloud and fog computing. | 14, 16, 12, 19, 44, 48, 60, 64, 61, 69 |
| Fog computing application with WSN. | 17, 18, 20, 21, 22, 23, 40, 42, 45, 47, 50 |
| Fog computing with WSN future directions. | 24, 26, 27, 29, 55, 66, 67, 69, 72, |
| WSN architectures for Fog and cloud computing. | 8, 9, 10, 12, 14, 15, 17, 19, 25, 26, 27, 40, 80, 82, 84 |
| Survey of Fog and cloud computing. | 28, 29, 30 |

**Table 4.** Search key strings.

| Database Sources | Search Strings |
|---|---|
| • Google Scholar<br>• ELSEVIER<br>• (Science Direct)<br>• Springer Link<br>• Research Gate<br>• IEEE<br>• Semantic<br>• Scholars | WSN AND (fog computing OR cloud computing) AND (relation of WSN with cloud OR Scope determination OR Requirements for a combination of WSN-Fog-Cloud) AND (Survey OR Review) AND (Architectures OR Modeling) AND (Challenges OR Issues OR Limitations OR Success Rate OR Failure Reasons) AND (Techniques OR Tool/s OR Methodologies). |
| | Fog AND (WSN OR Cloud) AND (Vs Development in combination) AND (Limitations OR Challenges OR Industrial Drawbacks) AND (Adoption Rate OR Success Rate) AND (Components OR Benefits OR Frameworks) AND (Project working Risks OR Project development Components) OR (Surveys). |
| | Cloud AND (Fog OR Combination OR WSN) AND (Definition OR Cloud-Fog-WSN OR Levels) AND (Enhancements OR Improvements OR IoT) OR (Success Factors OR Failure Reasons). |

The papers were gathered from Google Scholar and other search engines based on the inclusion and exclusion criteria. This criteria is applied in the bases of the key strings mentioned in Table 4. We only enlisted papers that have knowledge and data related to these aspects. We gathered 250 papers, and after applying the inclusion and exclusion criteria we shortlisted 100 papers for this study.

## 5. Comparison of Integration WSN with Cloud and Fog

The possibility for gathering information from WSN is high, though, required limits as far as capacity, and transforming energy. On the other hand, CC does not need any improvement for storage and transforming energy. Both the innovations are considered, after that WSN-CC and fog combination might evaluate a substantial number of issues we talked about in Table 5. We have included the framework which works for WSN with cloud and fog in terms of efficiency, working ability, and safety.

**Table 5.** Framework comparison of integration of WSN with cloud and Fog.

| Ref. No | Problem Addressed | Techniques | Strength | Weakness | Tools | Domain |
|---------|-------------------|------------|----------|----------|-------|--------|
| [1] | Integration get benefits and efficiency | Zigbee is used to route the packets and store the data | Customers can send and receive information to the server | Computation cost is high | Zigbee | WSN in CC |
| [2] | Minimize the battery power to maximize the lifetime of the node in WSN. | HEF (High Energy First) algorithm. Reduce the number of transmissions. | HEF algorithm lifetime can be bounded and by using a device cloud application. | These disorganized cluster heads could not maximize energy efficiency. | Leach | WSN in CC |
| [3] | Identification of the threatening and trustworthiness of WSN. | TPSS scheme consisting of TPSDT and PSS for WSN-MCC integration. | TPSS incorporates TPSDT and PSS to improve sensory data and the reliability of WSN. | Computational and energy cost is high | | WSN-MCC |
| [4] | Data shift from WSN to the cloud scientifically and economically. | Data of WSN moves through a gateway to the DPU. The DPU sends the data to DR according to storage format. | Data can be stored efficiently and in a systematic manner. | The Authors do not provide any mechanism to store and process the data. | Data Processing Unit (DPU), Pub/Sub Broker, Request Subscriber (RS), Identity and Access Management Unit (IAMU), and Data Repository (DR). | WSN in CC |
| [5] | Make sensor data available across the globe and reduce maintenance costs. | Authentication and access control are provided through Identity and Access Management Unit (IAMU). | Data accessed on a cloud through a secured Identity and Access Management Unit (IAMU) | No simulation results showed the accuracy of the framework. | Identity and Access Management Unit (IAMU). | WSN in CC |
| [6] | Minimize the storage requirements for sensor nodes and networks gateway. | Data transfer to mobile users in a rapid, reliable, and even more secure manner. | Minimizes the traffic overhead and bandwidth requirement and also framework predict the data trend with security. | Deploying more sensors to the area that mobile users are interested in | | WSN-MCC |
| [7] | Bandwidth quality suffers due to the collection of multimedia data. | Several proposed frameworks for integrating WSN-MCC are reviewed. | The optimal usage of WSNs can be managed and their status can be checked. | Due to compromising the possibility of WSNs, the data transferred through WSNs are not the data needed by mobile users. | | WSN-MCC |
| [8] | Facilitate connecting all components of the network. | Event matching algorithm called Statistical Group Index Matching (SGIM) which targets range predicate case. | Deliver published sensor data or events to appropriate users of cloud applications. | Need the computational CC model than the traditional HPC approaches | | WSN-MCC |
| [9] | limited resources of a sensor are the main challenge for deploying and operating WSNs. | Shortened the average end-to-end path length of packet transmission | the efficiency of sending operation is improved. | The purposed architecture is needed to be carefully managed. | | WSN in CC |

**Table 5.** *Cont.*

| Ref. No | Problem Addressed | Techniques | Strength | Weakness | Tools | Domain |
|---|---|---|---|---|---|---|
| [10] | Tracking of multiple targets using the sensor-cloud infrastructure. | Social-choice-based Dynamic Mapping Algorithm (S-DMA) within a sensor-cloud environment. | The S-DMA ensures the best possible allocation of sensors to targets. | If two adjacent sensor nodes are heterogeneous concerning their sensing types as multi-hop communication in such a scenario will require protocol standardization. | | WSN in CC |
| [11] | Critical issues that affect the use of sensory data and the reliability of WSN. | TPSDT (Time and Priority-based Selective Data Transmission) for WSN gateway to selectively transmit sensory data, PSS (Priority-based Sleep Scheduling) algorithm for WSN | TPSS incorporates TPSDT and PSS to improve both the use of sensory data and the reliability of WSN. | Load balancing and energy consumption are triggered by updating the virtual sensors in the cloud | | WSN-MCC |
| [12] | If nodes transmit the data continuously, the lifetime of the WSN will be short. | Location-based characteristics of mobile applications, as well as the energy concern of WSNs, are taken into account by the CLSS schemes. | Energy-saving. | Require protocol standardization. | | WSN-MCC |
| [13] | MCC applications are often utilized in a location-specific way. | Sleep scheduling in WSNs to support location-based mobile cloud applications. | Energy-saving. Satisfying the data requests of the mobile users. | Multi-hop communication in heterogeneous WSN needs protocol standardization. | | WSN-MCC |
| [14] | Lack of expertise and storage of gathered information not used adequately. | Sensor nodes are considered service providers and sink nodes are consumers. | Facilitate the shift of data from WSN to CC storage. | Space Constraint | | WSN in CC |
| [15] | Intensive sensor utilization in an SC (Sensor Cloud) that must have QoS guarantees. | The QoS-aware cloud model is provided | Real-time services for environmental sensing, monitoring, and process control systems | Require protocol standardization. | | WSN in CC |
| [16] | Trust and reputation calculation and management (ATRCM) framework for CC-WSN integration. | Propose an ATRCM structure that fulfills the three capacities: computing managing and selecting of CSP SNP and CSU | Authentication with trust and position calculation with the management of cloud service providers (CSPs) and sensor network suppliers (SNPs) | Space Constraint | | WSN in CC |
| [17] | Data management and communication over MSN (Mobile Sensor Network). | The gateway performs filtering, compression, monitoring, and prediction of the data. | The proposed technique minimizes the overhead, bandwidth requirement, and additional requirements for storage. | Deploying more sensors to the area | | WSN in CC |

Table 5. *Cont.*

| Ref. No | Problem Addressed | Techniques | Strength | Weakness | Tools | Domain |
|---|---|---|---|---|---|---|
| [18] | Data-centric centers look for collecting fine-detailed data about the domain area, and the sensor nodes through energy saving. | The proposed model assumes a linear distribution for the data and captures the correlation among the data by the line equation | The data was collected at the finest level while minimizing the energy consumption of sensor nodes. | Integrating WSN and CC will add extra load to the sensor nodes. | | WSN in CC |
| [47] | Due to the poor communication ability of WSN to cloud is challenging. | Approximation and detailed routing algorithm for sensors considering hops and energy consumption | Maximize throughput and minimize the latency with Energy Saving | In Fog layer sink act as a fog node that used more power. | DCF | IoT in FC |
| [48] | Dynamic Service Execution (DSE) problem in Fog to cloud scenarios | Proposing two basic resource allocation strategies, First-Fit and Random-Fit | Improve all key metrics Service Response Time, Power Consumption, Network Bandwidth, and Service Disruption Probability | | First-Fit and Random-Fit. | IoT in FC |
| [49] | Service delay in IoT and cloud | Proposed framework placed between cloud and IoT and design an analytical model for minimizing the service delay | An analytical model can support other policies of FC. | | Distributed and centralized mode of communication | FC |
| [50] | Energy efficiency, latency, performance, response time, and mobility are the problems in Fog | Develop a novel integrated fog cloud IoT (IFCIoT) architectural paradigm | Improve energy efficiency, reduce latency, increase performance and response time with mobility management | | Integrated fog cloud IoT (IFCIoT) | IoT in FC |
| [51] | Design goals and platforms are the challenges of a fog. | Present the design and implement a fog platform | Performance in terms of response improved | Present the prototype platform | Open Stack modules | IoT in FC |
| [52] | Cross-industry and cross-platform face identification. | Purposed a CC-based face resolution framework. | The proposed framework enables to provide the unified face identification for IoT applications. | Face identification performance is questionable if the face identifier templates are not periodically updated. | | IoT in cloud |
| [53] | Data transfer from IoT to cloud | Design a new protocol Peer assistant UDT-based Data Transfer Protocol (PaUDT) | Improve data transmission and congestion control. | | PaUDT protocol with P2P network. | IoT in FC |
| [54] | IP does not fill the need to integrate Sensor Networks with the Internet. | Information-Centric Networking (ICN)-based FOGG computing Gateway | Availability and extend the control of the sensor through intelligence data processing. | Uses Cases describe the functionality of FOGG. | Named Data Networking (NDN) and Information-Centric Networking (ICN) | WSN in Fog computing |

**Table 5.** *Cont.*

| Ref. No | Problem Addressed | Techniques | Strength | Weakness | Tools | Domain |
|---|---|---|---|---|---|---|
| [55] | In WSN, Cluster head selection is the main problem due to energy saving. | Proposed a FOG-supported sensor network using a new Stable Election protocol. | The proposed algorithm saves energy and maximizes the network life. | Use high-power sensors as a gateway. | New Stable Election Protocol(N-Sep) | WSN in Fog computing |
| [55] | Data delivery from WSN to cloud is the main problem of the integration of WSN and cloud. | Proposed a three-layer framework based on fog with multiple mobile sinks. | Multiple mobile fog nodes can cooperate to set up a mobile multi-input multi-output (MIMO). | Multiple mobile sinks use as the fog layer to bridge the communication gap b/w WSN and cloud. | Design a DDF (Data delivery with fog) algorithm. | WSN in Fog computing |
| [56] | Address the problem of localization of Ambient Assisted Living (AAL). | Present a fog-enabled WSN system with an Edge mining technique. | Due to the edge mining technique sensors send the data in as per a predefined format to the cloud. | Consider the constant speed of the user. | Iterative Edge Mining: IEM and Genetic Algorithm. | WSN in FC. |
| [57] | Application development for fog is challenging due to processing, heterogeneous, tightly coupled, and widely distributed devices. | Present a Distributed Dataflow (DDF) programming model. | The proposed model Utilizes computing infrastructures across the Fog and the cloud. | Distributed Dataflow (DDF) model defines their language. | Distributed Node-RED (D-NR). | IoT in FC |
| [58] | Unnecessary communication is a burden on the core network and the data center of the cloud. | Proposed a smart gateway for Data preprocessed and trimmed according to the format. | Based on the application feedback, Gateway must decide the time and type of data to be sent. | Only Suitable for mobile objects and large-scale IoT/WSN. | | WSN/IoT in FC |
| [59] | How to computation-intensive tasks offload effectively from resource-constrained devices. | Proposed a data acquisition mechanism for clustering WSNs. | Even in the case of data abnormality proposed approach enables reliable and efficient data acquisition. | | Unusual data filtering Algorithm and Suspicious data detection Algorithm. | WSN in FC |
| [60] | How can upload the data with different types and frequency on the cloud without extra burden on the core network and cloud? | Proposed fog-based efficient resource management framework. | Resources management through the probability of resource utilization and user characteristic. | For resource management, the characteristics of the user should be known. | | IoT in FC |
| [61] | Provide user services, reduce latency, and enabling real-time big data analytics. | Proposed an FC layer for WSN to manipulate more efficiently with different energy states. | Support real-time analytic process filters the data and sends it to the cloud. | Implement on a very small scale for an experiment. | Wireless sensor nodes working on shallow sleep, deep sleep, awake algorithms. | WSN in FC |
| [62] | How can meet the application deadline and energy saving during the communication of the cloud of things? | Develop a mathematical model of a three-tier cloud of things to access the applicability of the fog. | Cloud operations perform in the fog for saving energy and provide the data in real-time. | The cooperation of different entities of different tiers is not discussed. | | IoT in FC |

Then the assembled data are transferred to the cloud. These frameworks are demonstrated to be reliable, accessible, and extensible. This schema primarily centers on the use of data following that information may be transmitted to the cloud. Versatile clients prefer

that data, they did not need the raw data. Those mobile clients ask for the information starting with the cloud and the cloud performs information recommendations and predicts information after giving back these required data to the clients [61,62]. After predicting data, the cloud needs information characteristics that are more inclined to mobile users regarding that information. Cloud informs WSN to use this data to streamline the sending data by WSN.

*Resource Scheduling for WSN with Cloud and Fog*

FC brings organized resources close to the fundamental systems. FC expands the conventional CC worldview to the edge of the system to empower, refine, and for better applications or services. FC is a virtualized stage, which gives calculation, storing, and organizing services between the end nodes in IoT and traditional clouds [63].

With an expanding number of heterogeneous devices associated with IoT and creating information, it will be inconceivable for an independent IoT to effectively perform power and data transmission. IoT and distributed computing integration have been imagined to secure the data of the cloud [64], a circumstance when the cloud is associated with an IoT that produces interactive media information. Visual sensor networks (VSN) or closed-circuit television (CCTV) associated with the cloud can be cases of such a situation. Since interactive media content expends additional preparing power, storage room, and resource requirements, services in the cloud will unavoidable. Fog processing assumes an exceptionally fundamental part of the cloud [65].

Fog is actualized near the end clients. In this manner, FC gives a better nature of services in terms of system data transmission, control, utilization, throughput, and response time and it lessens the movement over the web. There are numerous resource assignment systems in CC. System resource allocation methodologies and how these techniques can be actualized in CC conditions are discussed in the study [66]. There are many planned calculations for resource provisioning. However, there is a need for a powerful resource provisioning methodology keeping in mind that the end goal is to satisfy the request of clients and limit the general cost for the clients and additionally for cloud servers. The primary target of resource provisioning calculation is to plan the virtual machines (VMs) on the server. There is little study on upgraded resource planning calculation, resource provisioning technique of the market planning with numerous Service Level Agreement (SLA) parameters, resource allotment control-based show, adaptable resource provisioning, blockage control resource allotment model and ask for forecast demonstrate.

Researchers have concentrated on two issues, provisioning and resource allocation in distributed computing [67]. First is the Hadoop Map Reduce (HMR) and its schedules, the second reservation issue is provisioning virtual machines to resources in the cloud. MapReduce is a programming model for the preparation of vast scale information and was initially created by Google and Hadoop given the execution of Map Reduce. There are three schedulers accessible: First In First Out (FIFO), reasonable scheduler, and limit scheduler. The second planning issue is the provisioning of VMs and the task of VMs on physical machines. Resource sharing planning is a fundamental issue in CC. Cloud service gives virtual resources to the effective framework.

There is an essential connection between the framework segment and its capacity utilization for execution in cloud conditions [68]. The energy utilization examination of cloud groups with the assistance of cloud group nodes has been proposed. Level 1: virtualization and physically, layer 2: fog sensors, servers, and gateways, level 3: supervision, levels 4: preprocessing and post-processing, level 5: storage and resource managing, level 6: safety, and level 7: applications are multiplatform of the fog computing standard architecture. All the above levels are displayed in Figure 5 as a multilayered fog architecture. These fog architecture levels are divided into categories according to the applications they are used for. The significance of each level is examined, as well as its applicability in diverse applications. The purpose of these levels will be to collaborate to transmit a task from an IoT to a fog node and finally to the cloud for accomplishment [69].

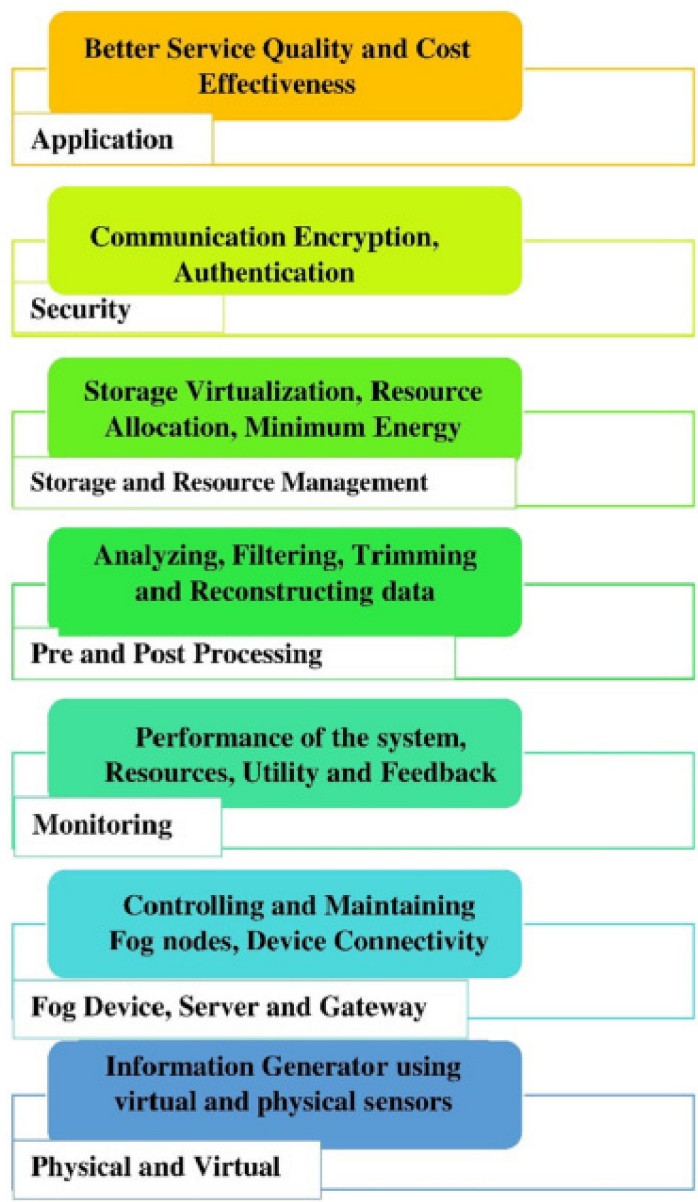

**Figure 5.** Architectural Solution for stated issues in Fog.

## 6. Gaps and Future Work

This work is done in two domains; the first is WSN integration with cloud and the second is an integration of WSN with fog. The integration of WSN to the cloud is an absolute technology and has many drawbacks as compared to WSN to Fog integration [70]. A dynamic, scaling, and extensible framework is required for integration from WSN to Fog. Data delivery from WSN to fog and vice versa is a challenging task and requires more attention from the researchers. Resource provision is also challenging due to the requirement of live streaming and monitoring with any delay. Energy efficiency is also a major factor that affects the credibility of WSN [70–72]. Furthermore, those encryption points exhibited in this paper might have been restricted to the main information and make it an open region for investigation [73–75].

For assignment mapping and planning, the creators have connected the ECO Map Algorithm (EMA) for special case jump grouped homogeneous WSN, however, its materialness again multi jump heterogeneous WSNs necessity should be reviewed further [76–78]. The location management problem in terms of QoS should be addressed. Resource pro-

visioning of the cloud to WSN is the main issue [67,79,80]. Furthermore, those execution parameters were chosen to streamline the main normal delay [81–83]. That could be a chance to be viewed as for the future worth of effort [76,84,85]. An integrated framework for WSN with fog is required to address all the above-mentioned issues [77,78,86,87].

We will deploy a lightweight integration framework of WSN and fog with load balancing and prediction of data type and next need of that data with the addition of artificial intelligence. We will compare our results with the latest framework of fog [88].

All the abbreviated acronyms mentioned in Table 6.

**Table 6.** Abbreviated acronyms.

| Acronyms | Meaning |
|----------|---------|
| CC | Cloud Computing |
| FC | Fog Computing |
| WSN | Wireless Sensor Network |
| LEACH | Low Energy Adaptive Clustering Hierarchy |
| FOGG | A Fog Computing Based Gateway to Integrate Sensor Networks to Internet |
| VM | Virtual Machines |
| IBM | International Business Machines |
| AAL | Ambient Assisted Living |
| LQRP | Link Quality Routing Protocol |
| HEF | High Energy First |
| TPSS | Time and Priority-based Selective Data Transmission |
| IAMU | Identity and Access Management Unit |
| CCTV | Closed-Circuit Television |
| VSN | Visual Sensor Network |

## 7. Conclusions

CC will be an innovative standard that gives convenience; the on-demand system gets an imparted pool of configurable computing resources. That might have a chance to be a quick provision of computing resources and settle for insignificant low-cost utilization of service providers. The perfect coordination of WSN includes a vast number of low cost, low control multi-working nodes with CC, which is another rising area that gives a strong and versatile foundation for a few requirements. In this paper, we surveyed the requirements, tests, and results identified for coordination between WSN and cloud. Furthermore, issues like security, protection, and coordination are still needed to be attended to. The scalability, process, delay constraint, routing, and heterogeneity are other issues that need to be addressed.

Executing FC on an ad hoc system helps to decide immediately before any problem arises. FC moves the edge of the system with the least idleness, less processing, and system service benefits. This kind of framework can be utilized as a part of health, augmented reality, and in numerous ongoing Internet of Things (IoT) applications like visual security, etc.

**Author Contributions:** Conceptualization, Data Creation, Investigation, Resources, Validation, Writing—review & editing, Formal analysis Q.S.; Project administration, Resources, Supervision M.S.; funding acquisition A.G.; Visualization M.A.K.; methodology, Formal analysis, Validation, Writing—review & editing S.A.B. All authors have read and agreed to the published version of the manuscript.

**Funding:** This is partially collaboration work with University Malaysia Sabah, Malaysia.

**Conflicts of Interest:** The authors declare no conflict of interest.

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
