# Peer review of "An Analytical Survey of WSNs Integration with Cloud and Fog Computing"

_electronics, doi:10.3390/electronics10212625_

Round 1
Reviewer 1 Report
In this manuscript the Authors propose a survey focusing on the integration of Wireless Sensor Networks (WSNs) with Cloud and Fog computing (CC/FC). Several studies are considered with the aim of studying the various frameworks integrating WSN and CC/FC. The manuscript introduces the reader to the needed background, then different data sources are used to extrapolate studies regarding the context.
The manuscript expresses potential: the integration of WSNs with other methodologies and techniques, such as CC and FC, is a extremely hot topic. However, in the current state, the manuscript misses its expressed intentions. In particular, the presentation lacks details and the considered studies are not scrutinized enough.
The manuscript is interesting, thus I suggest the Authors to address the following aspects to improve it:
- The Introduction, and in particular the last part of it, should be made clearer; in particular, the last part is confusing and does not provide enough information on the rationale behind the paper (e.g., why the study therein proposed is useful), the gap the paper is filling and the main contributions. I suggest the Authors to improve it or, at least, to blend it with the Section 2.
- Few recent studies related to the integration between WSN and cloud computing are missing. In particular, the Authors might take into consideration the following related pointers [1,2].
- As data sources, the Authors take into account (citing from the paper) "Google Scholar, Scopus and Google". Regarding the last one (Google), is the data filtered? If so, what are the considered sources? (Not including the ones in Table 4)
- The framework comparison should not appear in a table. Instead, it should be discussed through the whole text and the table should only provide a bird's eye-view of the considered content.
Also, there are few minor notes:
- Acronyms in text should be redefined when used for the first time (e.g., Fog Computing).
- Several typos and repetition of thoughts are present (e.g., compuTTing), I suggest the Authors to proofread the manuscript.
- The overall presentation, and the english construction of the paper should be improved to be acceptable.
Author Response
We have accomodated all the suggested changes.

Reviewer 2 Report
In order to overcome the WSN`s limitations this paper propose o solution for integrating the WSNs with the cloud computing and with the Fog computing. The integration is done in two ways: the WSN integration with cloud and the integration of WSN with fog. The integration of WSN to the cloud is an absolute technology and has many drawbacks in comparison to WSN to Fog integration.
The authors analysed the requirements, tests, and results identified for WSN and cloud coordination, as well as the security, protection and coordination issues, the scalability, process, the delay constraint, routing, and heterogeneity issues that need to be addressed.
The strategy for study methodology relies on search engines such as Google Scholar, Scopus and Google and a systematic literature review was analyzed for identifying numerous elements that are significant in defining the scope of study.
Weakness
- The related work section does not clearly show the contributions listed to whom they belong. The presentation is very ambiguous
- There are acronyms used in the paper that are not explained (eg. FOGG, ICN, LEACH …)
- The paper presents a literature survey, but it would be useful to propose a detailed architectural solution for the identified problem
- The paper is oriented on the identification and systematization of bibliographic sources that treat the identified problems of interest, but do not bring new technical solutions.
Author Response
we covered all the suggested changes.

Round 2
Reviewer 1 Report
The Authors successfully addressed my concerns, except the one on related studies to the integration between WSN and cloud computing. I acknowledge that it was my fault because I reported to the Authors to consider to include two pointers, but I actually did not provide them. The pointers are doi.org/10.1016/j.inffus.2018.11.010 and doi.org/10.1016/j.pmcj.2020.101223.
Author Response
File is attached

Reviewer 2 Report
The authors have responded to my observations: the acronyms in text were redefined, the paper was completed with Fig.4 and the explanations associated, the framework comparison from table 5 is discussed in some explanations. It would be useful to have given a schematic recommendation related to the comparative framework ...
Author Response
File is attached
